# Sex disparities in adult obesity prevalence across 47 African countries: A cross-sectional descriptive study

Augustus Osborne◉*

Institute for Development, Western Area, Freetown, Sierra Leone

* augustusosborne2@gmail.com

## Abstract

### Background

Obesity is a growing public health concern in Africa, with evidence suggesting pronounced disparities between men and women. Understanding the prevalence and sex differences in obesity across African countries is critical for informing equity-focused interventions. This study examined the prevalence of adult obesity (body mass index [BMI] ≥30) across 47 African countries using sex-disaggregated data to assess the magnitude of sex disparities in obesity prevalence as a basis for equity-focused public health interventions.

### Methods

This study is a cross-sectional descriptive study using nationally representative, sex-disaggregated data from 47 African countries, primarily for the year 2022, obtained from the World Health Organization Health Equity Assessment Toolkit (HEAT). Adults aged 18 years and older who had recent, available data were included in the study. The primary outcome was the country-level prevalence of adult obesity (body mass index [BMI] ≥30) disaggregated by sex. The secondary outcome was the absolute difference in obesity prevalence between women and men within each country.

### Results

Obesity prevalence among adults varied widely across African countries. South Africa had the highest prevalence at 30.8%, followed by Eswatini (30.1%) and Seychelles (29.4%), while Ethiopia (2.8%), Madagascar (4.3%), and Eritrea (4.8%) reported the lowest rates. Across nearly all countries, women exhibited considerably higher obesity prevalence than men. In South Africa, female obesity prevalence was 45.8%, compared to 13.9% in men a difference of 31.8 percentage points. Large sex disparities were also observed in Eswatini (difference: 28.8), Mauritania (26.1), Lesotho

provided the original author and source are credited.

**Data availability statement:** Description and source of data This study used third-party, publicly accessible, country-level, sex-disaggregated estimates of adult obesity (BMI ≥30) obtained from the World Health Organization's Health Equity Assessment Toolkit (HEAT), built-in database edition, Version 6.0 (2024), which draws from the WHO Health Inequality Monitor data repository and the WHO Global Health Observatory (GHO). The specific indicator used was "Obesity prevalence among adults, BMI≥30 (age-standardized) (%)," disaggregated by sex. The HEAT platform and the underlying data repository are available at: • WHO Health Equity Assessment Toolkit (HEAT): https://www.who.int/data/inequality-monitor/assessment_toolkit • WHO Health Inequality Monitor – Data repository (latest version): https://www.who.int/data/inequality-monitor/data • WHO Global Health Observatory: https://www.who.int/data/gho Access instructions Interested researchers can access the same data without restriction by visiting the links above. Within HEAT, select "Compare inequality," choose the indicator "Obesity prevalence among adults, BMI≥30 (age-standardized) (%)", select "Sex" as the dimension, and choose the country and year (primarily 2022 for this analysis). Data and metadata can be exported directly from HEAT. The GHO provides methodological notes and indicator definitions. Permissions and privileges The author did not receive special access privileges; all data are publicly available. HEAT and the WHO repositories permit users to download and use country-level estimates with appropriate citation of WHO/HEAT and the data sources. Contact information For questions about accessing or using the WHO HEAT or WHO Health Inequality Monitor data, please contact: • WHO Health Inequality Monitoring team via the HEAT/Health Inequality Monitor pages above • General WHO data enquiries: data@who.int Additional documentation is available on the HEAT website and the GHO indicator pages.

**Funding:** The author(s) received no specific funding for this work.

**Competing interests:** The authors have declared that no competing interests exist.

(24.2), Equatorial Guinea (21.8), and Seychelles (19.4). In contrast, only a few countries, such as Burundi, Chad, and Madagascar, showed negligible or slightly higher obesity prevalence among men.

## Conclusion

The findings reveal that adult obesity is a pronounced and growing public health concern in Africa, with marked heterogeneity in prevalence between countries and an overwhelming burden among women. These disparities reflect the sociocultural, economic, and biological factors influencing obesity risk, including urbanization, dietary transitions, and gender norms. The pronounced sex disparity points to the need for context-specific, sex-sensitive interventions and policies to effectively address obesity and its health consequences. Policymakers and health practitioners should prioritize multisectoral strategies that promote healthy diets and physical activity and address the unique barriers faced by women. Enhanced surveillance and research are also needed to further elucidate the determinants of obesity and to monitor progress toward reducing health inequities across the continent, including fiscal, regulatory, and community-based actions tailored to women's needs.

## Introduction

Obesity is a major public health challenge that contributes substantially to global morbidity and mortality through increased risks of cardiovascular disease, type 2 diabetes, selected cancers, and musculoskeletal disorders [1–5]. Global prevalence has nearly tripled since 1975, with more than 650 million adults now living with obesity (BMI ≥ 30) [1,6]. Although once perceived as concentrated in high-income settings, obesity is rising rapidly in low- and middle-income countries (LMICs), driven by shifts in diets and physical activity within the broader nutrition transition [4,7].

In Africa, adult obesity is increasing in many countries, often alongside persistent undernutrition, creating a double burden with serious health and economic implications [5]. Urbanization, economic transitions, and changes in food environments are key contextual factors, while limited preventive infrastructures and uneven policy responses compound the challenge [5]. Understanding the distribution and drivers of obesity in African contexts is therefore essential for guiding effective, equitable responses.

Across much of the continent, women have higher obesity prevalence than men, reflecting interacting sociocultural, economic, environmental, and biological factors [8–11]. In several countries, more than one in five adult women are affected, while male prevalence is markedly lower [9,10]. These sex differences have important equity and policy implications, particularly where gender norms, caregiving roles, and barriers to physical activity and healthy eating intersect [12–16].

Despite growing attention to obesity in Africa, comparable, up-to-date, sex-disaggregated estimates remain limited. Existing studies often focus on single

countries or subregions, use heterogeneous methods, or draw on data collected at different times, complicating cross-country comparisons [14–18]. These variations hinder a clear understanding of the magnitude and patterning of sex disparities across the continent and limit the ability to prioritize context-appropriate interventions.

This study provides comparable, sex-disaggregated estimates of adult obesity prevalence (BMI ≥ 30) across 47 African countries using the World Health Organization's Health Equity Assessment Toolkit (HEAT) [19]. It describe country-level prevalence, quantify female–male differences, and identify settings with the largest disparities to inform equity-focused interventions. By leveraging a standardized and validated platform, the study aims to strengthen the evidence base for policy and practice and to align with global agendas to reduce non-communicable diseases, including the WHO Global Action Plan for the Prevention and Control of NCDs and the Sustainable Development Goals (notably SDG 3).

## Methods

### Study design and period

This study adopted a cross-sectional design to estimate the sex-specific prevalence of adult obesity (BMI ≥ 30) across 47 African countries. The cross-sectional approach was selected as it allows for the assessment of health outcomes and disparities at a specific point, offering a snapshot of obesity patterns and sex differences during the study period. The analysis focused on the most recent available data for each country, primarily from the year 2022, as compiled by the World Health Organization (WHO). Countries were included based on the availability of up-to-date and comparable obesity data in the WHO Health Equity Assessment Toolkit (HEAT). Countries lacking sufficient or recent data were excluded to ensure the robustness and comparability of the results.

### Data source

The study used nationally representative, sex-disaggregated data from the WHO Health Equity Assessment Toolkit (HEAT) [19], an interactive software platform for exploring health inequalities drawing on WHO-approved survey sources (e.g., DHS, STEPS). HEAT is available at: https://www.who.int/data/inequality-monitor/assessment_toolkit. The study also cross-referenced metadata from the WHO Global Health Observatory (GHO): https://www.who.int/data/gho. These surveys employ robust sampling techniques to ensure data accuracy and representativeness, making them reliable for monitoring health indicators like obesity prevalence.

The study included African countries with adult (≥18 years) obesity (BMI ≥ 30) estimates disaggregated by sex and available in HEAT for the most recent year (primarily 2022). Of 54 countries in the African Region, 47 met these criteria. Countries excluded due to missing or non-recent sex-disaggregated estimates in HEAT include Egypt, Libya, Morocco, Somalia, Sudan, Tunisia, Djibouti, and Western Sahara. The included countries correspond to WHO African region members for which comparable estimates were available during the study period.

### Outcome measure and dimension of disparity

The principal outcome of interest was the prevalence of obesity, defined as the proportion of adults (aged 18 years and above) with a body mass index (BMI) of 30 or higher at the time of the survey. The data were disaggregated by sex (male and female) to enable a detailed assessment of sex-based differences in obesity prevalence.

Focusing on sex as an equity dimension is essential for understanding how obesity is distributed between men and women and for identifying disparities that may warrant targeted interventions. The study reports the absolute difference (D) in percentage points between female and male prevalence (female minus male). These descriptive measures are accompanied by 95% CIs from HEAT.

## Data analysis

All analyses were performed within the WHO HEAT platform, which offers a streamlined interface for exploring and comparing health inequalities. HEAT includes several analytical modules; for this study, the "compare inequality" module was used to generate sex-disaggregated estimates of adult obesity prevalence for each of the 47 countries. The analytical process involved the following steps: accessing the WHO HEAT software via the WHO website and selecting the "compare inequality" module. Choosing the indicator "obesity prevalence (BMI ≥30) among adults" from the available health indicators. Selecting sex as the dimension of inequality to disaggregate the data. Extracting sex-specific prevalence estimates for each country, along with the absolute difference in prevalence between males and females. A negative difference indicates higher obesity prevalence among women compared to men. Exporting sex-disaggregated estimates and 95% CIs from HEAT. The platform applies standardized statistical methods, such as sample weighting and stratification, to produce nationally representative estimates. Countries were ordered in descending prevalence. I summarized disparities using the median and interquartile range (IQR) of D and counted countries with higher female vs. male prevalence. Results are provided in tabular formats to facilitate interpretation and highlight the magnitude of sex disparities in obesity prevalence across countries. Data were exported from HEAT and organized into tables for inclusion in the manuscript.

Countries or subgroups without reliable, sex-disaggregated obesity prevalence data were excluded from the analysis. No imputation was performed, as the HEAT platform uses only validated, pre-analyzed data from WHO-endorsed sources. This approach ensures the analysis is based on high-quality and complete data, minimizing the risk of bias from missing information. The HEAT platform ensures comparability and standardization by harmonizing indicator definitions and survey methodologies and applying rigorous quality control procedures. This includes adjusting for sampling design and response rates to ensure estimates are representative and comparable across different settings.

No additional statistical analyses or modeling were conducted outside of the HEAT environment. All estimates and calculations are based on the WHO's standardized procedures, ensuring transparency, reproducibility, and alignment with global best practices for health inequality monitoring. By utilizing the HEAT toolkit, this study benefits from the WHO's methodological expertise, providing reliable and comparable estimates of sex disparities in adult obesity across Africa.

## Ethics approval and considerations

This study used publicly available, anonymized data from the World Health Organization Global Health Observatory and the Health Equity Assessment Toolkit. As no primary data were collected and no individual-level identifiers were used, ethical approval and informed consent were not required.

## Results

### Prevalence of obesity among adults in Africa

Overall prevalence ranged from 30.8% in South Africa, 30.1% in Eswatini, and 29.4% in Seychelles to 2.8% in Ethiopia, 4.3% in Madagascar, and 4.8% in Eritrea (Table 1). The top 10 countries by prevalence were South Africa, Eswatini, Seychelles, Algeria, Mauritania, Lesotho, Gabon, Botswana, Mauritius, and Namibia. The bottom 10 included Ethiopia, Madagascar, Eritrea, Rwanda, Burundi, Niger, Central African Republic, Malawi, Democratic Republic of Congo, and Sierra Leone.

### Sex- disaggregated obesity prevalence among adults in Africa

In nearly all countries, female prevalence exceeded male prevalence (Table 2). For example, in South Africa, female prevalence was 45.8% (95% CI: 41.5–50.3) versus 13.9% (11.1–17.1) in men; in Eswatini, 43.9% (37.2–50.7) versus 15.0% (10.4–20.4); and in Seychelles, 39.6% (31.8–47.4) versus 20.2% (13.7–27.5). Exceptions included Burundi, Chad, and Madagascar, where male prevalence was similar to or slightly higher than female prevalence.

**Table 1. Obesity prevalence among adults in African countries (descending order).**

| Rank | Country | Obesity Prevalence (%) |
|---|---|---|
| 1 | South Africa | 30.8 |
| 2 | Eswatini | 30.1 |
| 3 | Seychelles | 29.4 |
| 4 | Algeria | 23.8 |
| 5 | Mauritania | 22.7 |
| 6 | Lesotho | 21.0 |
| 7 | Gabon | 21.0 |
| 8 | Mauritius | 19.2 |
| 9 | Botswana | 18.3 |
| 10 | Equatorial Guinea | 17.7 |
| 11 | Liberia | 17.0 |
| 12 | Namibia | 17.0 |
| 13 | Comoros | 16.3 |
| 14 | Sao Tome and Principe | 16.5 |
| 15 | Cape Verde | 15.8 |
| 16 | Cameroon | 14.9 |
| 17 | Gambia | 14.9 |
| 18 | Zimbabwe | 14.2 |
| 19 | Ghana | 12.9 |
| 20 | United Republic of Tanzania | 12.6 |
| 21 | Kenya | 12.4 |
| 22 | Nigeria | 12.4 |
| 23 | Benin | 11.2 |
| 24 | Zambia | 11.1 |
| 25 | Mali | 11.4 |
| 26 | Angola | 11.5 |
| 27 | Guinea-Bissau | 11.5 |
| 28 | Togo | 11.6 |
| 29 | Cote d'Ivoire | 11.6 |
| 30 | Mozambique | 10.3 |
| 31 | Senegal | 10.2 |
| 32 | Guinea | 9.5 |
| 33 | Central African Republic | 9.3 |
| 34 | South Sudan | 8.6 |
| 35 | Congo | 8.5 |
| 36 | Uganda | 7.9 |
| 37 | Malawi | 7.7 |
| 38 | Sierra Leone | 7.1 |
| 39 | Burkina Faso | 6.7 |
| 40 | Chad | 6.7 |
| 41 | Democratic Republic of the Congo | 6.6 |
| 42 | Niger | 6.0 |
| 43 | Burundi | 5.0 |
| 44 | Rwanda | 4.9 |
| 45 | Eritrea | 4.8 |
| 46 | Madagascar | 4.3 |
| 47 | Ethiopia | 2.8 |

**Table 2. Sex-disaggregated obesity prevalence among adults in Africa.**

| Country | Female (%) | Female (Lower–Upper) | Male (%) | Male (Lower–Upper) |
|---|---|---|---|---|
| South Africa | 45.8 | 41.5–50.3 | 13.9 | 11.1–17.1 |
| Eswatini | 43.9 | 37.2–50.7 | 15.0 | 10.4–20.4 |
| Seychelles | 39.6 | 31.8–47.4 | 20.2 | 13.7–27.5 |
| Mauritania | 35.0 | 30.2–40.1 | 8.9 | 4.1–15.9 |
| Algeria | 32.3 | 27.1–37.7 | 15.5 | 11.6–20.0 |
| Lesotho | 32.7 | 26.9–38.8 | 8.4 | 5.5–12.0 |
| Gabon | 31.7 | 27.0–36.4 | 11.0 | 5.8–17.8 |
| Equatorial Guinea | 29.6 | 21.9–37.8 | 7.8 | 2.7–16.2 |
| Botswana | 27.2 | 21.4–33.6 | 8.4 | 5.1–12.8 |
| Mauritius | 25.3 | 20.0–30.7 | 12.9 | 9.0–17.5 |
| Comoros | 24.5 | 18.9–30.9 | 8.0 | 4.7–12.6 |
| Sao Tome and Principe | 23.3 | 19.3–27.7 | 9.7 | 6.9–12.8 |
| Cape Verde | 22.9 | 19.2–26.7 | 8.4 | 6.2–11.0 |
| Namibia | 22.8 | 17.2–28.9 | 10.2 | 5.4–16.3 |
| Liberia | 21.0 | 18.3–23.7 | 12.8 | 10.3–15.6 |
| Zimbabwe | 20.9 | 16.4–26.1 | 5.5 | 3.4–8.0 |
| Ghana | 20.3 | 17.0–23.8 | 5.1 | 3.6–7.1 |
| Gambia | 20.0 | 16.8–23.4 | 9.5 | 5.5–14.9 |
| Cameroon | 20.4 | 17.0–23.9 | 9.3 | 6.1–13.0 |
| United Republic of Tanzania | 18.3 | 14.5–22.4 | 6.5 | 3.9–9.9 |
| Kenya | 18.5 | 15.9–21.4 | 6.0 | 4.6–7.7 |
| Togo | 17.4 | 14.7–20.3 | 5.6 | 4.2–7.3 |
| Nigeria | 16.5 | 13.5–19.8 | 8.2 | 5.9–10.8 |
| Angola | 16.3 | 9.9–24.2 | 6.3 | 2.9–11.3 |
| Zambia | 16.4 | 12.9–20.2 | 5.1 | 3.2–7.4 |
| Senegal | 15.5 | 11.8–19.6 | 3.9 | 2.3–6.0 |
| Cote d'Ivoire | 15.7 | 12.8–19.0 | 7.7 | 4.1–12.6 |
| Benin | 15.0 | 12.0–18.2 | 7.1 | 4.6–10.4 |
| Guinea-Bissau | 14.8 | 10.0–20.7 | 7.6 | 2.9–15.4 |
| Mozambique | 13.7 | 10.0–18.0 | 6.2 | 3.7–9.6 |
| Mali | 12.9 | 10.1–16.2 | 9.9 | 5.9–15.1 |
| Congo | 12.8 | 8.8–17.3 | 4.2 | 2.1–7.3 |
| Central African Republic | 12.3 | 8.3–17.2 | 6.1 | 3.7–9.3 |
| Uganda | 11.3 | 9.5–13.4 | 4.1 | 3.1–5.3 |
| Malawi | 11.8 | 9.2–14.6 | 3.1 | 1.9–4.7 |
| South Sudan | 11.8 | 6.8–18.4 | 5.1 | 2.4–9.2 |
| Sierra Leone | 11.4 | 8.9–14.4 | 2.7 | 1.8–3.8 |
| Guinea | 12.7 | 9.8–16.3 | 5.9 | 3.0–10.1 |
| Chad | 6.0 | 4.0–8.7 | 7.4 | 3.5–13.1 |
| Niger | 7.6 | 6.0–9.5 | 4.4 | 3.1–6.0 |
| Rwanda | 7.7 | 6.4–9.3 | 1.8 | 1.2–2.6 |
| Burkina Faso | 9.3 | 7.7–11.0 | 4.0 | 2.8–5.5 |
| Democratic Republic of Congo | 9.0 | 6.2–12.1 | 4.2 | 2.3–6.9 |
| Madagascar | 4.1 | 3.0–5.5 | 4.5 | 2.0–8.3 |
| Burundi | 4.0 | 2.6–5.8 | 6.1 | 2.1–12.9 |
| Eritrea | 6.6 | 3.9–10.3 | 2.8 | 1.3–5.2 |
| Ethiopia | 4.5 | 3.2–6.1 | 1.1 | 0.6–1.7 |

## Disparity in s sex disaggregated obesity prevalence among adults in Africa

Table 3 reveals substantial disparity in obesity prevalence between adult females and males across African countries, with most nations showing markedly higher rates among women. South Africa exhibits the greatest disparity (D = 31.8), followed by Eswatini, Mauritania, and Lesotho, all with differences exceeding 20 percentage points. Only a few countries, such as Madagascar, Chad, and Burundi, have negative values, indicating higher obesity prevalence among men.

When grouped by World Bank income level, upper-middle-income countries tended to have higher adult obesity prevalence and larger female–male gaps than lower-income peers. For example, countries such as South Africa, Botswana, Gabon, and Seychelles (upper-middle income) cluster toward the upper end of both overall prevalence and D, whereas many low-income countries (e.g., Ethiopia, Niger, Madagascar, and Burundi) lie at the lower end. The median female–male difference was 11.6 percentage points (IQR: 7.5–18.8; range: −2.1 to 31.8) (Table 4).

## Discussion

This analysis reveals wide heterogeneity in adult obesity prevalence across African countries, with a consistent pattern of higher prevalence among women. These patterns align with the continent's ongoing nutrition and epidemiologic transition and highlight the need for context-sensitive, equity-focused responses.

In the African context, higher-income and higher-GDP settings tend to exhibit more advanced nutrition transitions and more intensive commercial influences, which help explain the larger female–male gaps and higher overall obesity prevalence observed in these countries [20]. Upper-middle-income economies typically have greater penetration of ultra-processed foods and SSBs through modern retail (supermarkets, convenience chains) and quick-service restaurants, alongside denser and more sophisticated marketing often targeted at women and youth via television, social media, and point-of-sale promotions [21,22]. These markets also feature larger price differentials that make energy-dense products relatively affordable compared with minimally processed foods. At the same time, rapid motorization and land-use patterns favouring car-dependent growth reduce routine physical activity, while safe, accessible infrastructure for walking, cycling, and public transport remains uneven. Where fiscal and regulatory policies are adopted but set at low rates, contain loopholes, or lack enforcement, their capacity to counter these commercial pressures is limited [23–25]. By contrast, many low-income settings have lower market penetration of ultra-processed products, smaller formal retail footprints, and more active travel out of necessity, which may partially constrain obesity levels though these contexts face the risk of rapid change as incomes rise and retail and marketing expand [26]. Taken together, these structural and commercial dynamics align with the socioeconomic gradient we observe and underscore the need for stronger, well-enforced packages of fiscal, regulatory, urban design, and food-environment policies tailored to countries at different stages of economic development.

While the focus was sex disparities due to data availability and scope, equity concerns extend to socioeconomic status, education, and place of residence. In many contexts, urban populations and higher-income groups initially experience higher obesity prevalence as food environments shift; over time, burdens can diffuse and increasingly affect lower-income groups as ultra-processed foods become more affordable and physical activity opportunities remain uneven [27]. The HEAT platform supports disaggregation by multiple equity dimensions (e.g., wealth, education, residence), offering a path for future analyses to identify intersectional vulnerabilities for example, women in low-income urban neighbourhoods who face concentrated marketing of unhealthy foods, limited access to affordable healthy options, and constrained opportunities for safe physical activity. Policymaking should therefore combine sex-sensitive strategies with pro-poor design, geographic targeting, and protection of children and adolescents.

The patterns observed in Africa echo experiences in other LMIC regions undergoing rapid nutrition transition. In parts of Latin America, the Middle East and North Africa, and Small Island Developing States in the Pacific, obesity prevalence has risen alongside expanding ultra-processed food markets, urbanization, and changing social norms [28]. Policy responses vary in intensity: Latin America has pioneered large front-of-pack warning labels and marketing restrictions (e.g., Chile, Mexico) [29], while the Gulf region has adopted SSB and energy drink taxes and, in some cases, front-of-pack measures

**Table 3. Disparity in sex-disaggregated obesity prevalence among adults in Africa.**

| Country | Difference (D) |
|---|---|
| South Africa | 31.8 |
| Eswatini | 28.8 |
| Mauritania | 26.1 |
| Lesotho | 24.2 |
| Equatorial Guinea | 21.8 |
| Gabon | 20.7 |
| Seychelles | 19.4 |
| Botswana | 18.8 |
| Comoros | 16.5 |
| Algeria | 16.8 |
| Zimbabwe | 15.4 |
| Ghana | 15.2 |
| Cabo Verde | 14.5 |
| Sao Tome and Principe | 13.6 |
| Namibia | 12.6 |
| Mauritius | 12.4 |
| Kenya | 12.5 |
| Senegal | 11.6 |
| United Republic of Tanzania | 11.8 |
| Togo | 11.8 |
| Zambia | 11.3 |
| Cameroon | 11.1 |
| Gambia | 10.5 |
| Angola | 10.1 |
| Malawi | 8.7 |
| Sierra Leone | 8.8 |
| Nigeria | 8.3 |
| Liberia | 8.2 |
| Côte d'Ivoire | 8.0 |
| Benin | 7.9 |
| Mozambique | 7.5 |
| Guinea-Bissau | 7.3 |
| Uganda | 7.2 |
| Guinea | 6.8 |
| South Sudan | 6.6 |
| Rwanda | 6.0 |
| Central African Republic | 6.2 |
| Congo | 8.6 |
| Burkina Faso | 5.3 |
| Democratic Republic of Congo | 4.8 |
| Eritrea | 3.8 |
| Niger | 3.3 |
| Mali | 3.0 |
| Ethiopia | 3.5 |
| Madagascar | −0.4 |

*(Continued)*

**Table 3.** (Continued)

| Country | Difference (D) |
|---|---|
| Chad | −1.4 |
| Burundi | −2.1 |

Difference (D) is defined as the absolute difference in obesity prevalence between adult females and adult males in each country. Positive values indicate higher prevalence among females; negative values indicate higher prevalence among males.

**Table 4.** Female–male difference in adult obesity prevalence (D, Percentage Points) by World Bank income group.

| Income Group | Number of Countries (n) | Median D | IQR (25th–75th pct) | Range (min, max) | Countries with D ≥ 20 pp (count) |
|---|---|---|---|---|---|
| Upper-middle income | 8 | 20.0 | 18.8–21.8 | 12.4, 31.8 | 5 |
| Lower-middle income | 21 | 11.6 | 8.0–15.4 | −0.4, 28.8 | 1 |
| Low income | 18 | 7.2 | 5.3–11.1 | −2.1, 16.8 | 0 |
| All countries | 47 | 11.6 | 7.5–18.8 | −2.1, 31.8 | 6 |

[30]. These international experiences underscore that comprehensive packages combining fiscal and regulatory actions with healthier food procurement, school-based policies, and urban design are more likely to be effective than isolated measures. They also illustrate that early adoption and sustained enforcement are critical to impact.

Biological factors including hormonal milieu and pregnancy-related weight changes may contribute to higher obesity prevalence among women within countries [31]. However, these factors are universal and cannot account for the pronounced inter-country heterogeneity observed. Differences between countries are more plausibly driven by sociocultural norms (including body image ideals), economic conditions, the structure of food environments, marketing intensity, and the presence or absence of effective fiscal and regulatory policies. The study interpretation therefore places biological explanations in appropriate context: relevant to within-country sex differences, but insufficient to explain between-country variation.

## Policy and practice implications

These findings reinforce the case for comprehensive, equity-focused obesity prevention and control strategies that prioritize implementing or strengthening SSB taxes at levels sufficient to change prices and demand, indexing to inflation, and considering extensions to high-sugar foods, earmarking revenues for prevention and treatment. Adopting front-of-pack warning labels; restricting marketing of unhealthy foods and beverages to children; establishing nutrition standards for schools and public procurement; monitoring and enforcing compliance. Supporting healthy retail in underserved areas; regulating food availability around schools; leveraging social protection to improve diet quality. Expanding safe, accessible infrastructure for walking and cycling; integrating physical activity considerations into city planning with attention to safety and equity. Integrating obesity prevention and management into primary care and maternal and women's health services; training providers in brief interventions and referral pathways. Co-designing interventions with affected groups to ensure cultural relevance and uptake, particularly among women and low-income urban residents.

## Strengths, limitations, and future directions

A strength of this work is the use of standardized, sex-disaggregated estimates across many African countries, which enhances comparability for regional policy dialogue. However, HEAT compiles secondary data with variable survey timing, instruments, and sampling across countries, which can affect cross-country comparability despite WHO standardization. Future analyses should leverage the full capabilities of HEAT to examine socioeconomic and geographic inequalities,

assess time trends as new data become available, and evaluate associations with policy adoption (e.g., SSB taxes, labelling) and urbanization dynamics. Mixed-methods research that combines quantitative monitoring with qualitative inquiry into commercial and sociocultural drivers will be essential to design and scale equitable interventions.

## Conclusions

This study provides a detailed picture of the landscape of adult obesity in Africa, revealing substantial variation in prevalence both across countries and between sexes. The findings highlight that women are disproportionately affected by obesity in nearly all African countries, with the sex variation being particularly wide in Southern and Western Africa. These patterns reflect the combined influence of sociocultural, economic, and biological factors and underscore the need for targeted, context-specific interventions. Addressing the obesity epidemic in Africa will require coordinated action across sectors, with particular attention to the unique challenges faced by women. Early and sustained efforts to promote healthy lifestyles and environments are essential to curbing the rising tide of obesity and its associated health consequences on the continent.

## Author contributions

**Conceptualization:** Augustus Osborne.

**Data curation:** Augustus Osborne.

**Investigation:** Augustus Osborne.

**Methodology:** Augustus Osborne.

**Supervision:** Augustus Osborne.

**Validation:** Augustus Osborne.

**Writing – original draft:** Augustus Osborne.

**Writing – review & editing:** Augustus Osborne.

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
