## [Decision Letter · Decision Letter 0]

8 Oct 2025

Dear Dr. Osborne,

Thank you for submitting your manuscript to PLOS ONE. After careful consideration, we feel that it has merit but does not fully meet PLOS ONE’s publication criteria as it currently stands. Therefore, we invite you to submit a revised version of the manuscript that addresses the points raised during the review process.

We look forward to receiving your revised manuscript.

Kind regards,

Olushayo Oluseun Olu

Academic Editor

PLOS ONE

Additional Editor Comments:

Please ensure that you address all the comments of both reviewers and provide a point-by-point explanation of the changes you made to the revised manuscript. Additionally, please ensure that all changes are marked in track change in the revised manuscript. Thank you.

Reviewers' comments:

Reviewer's Responses to Questions

**Comments to the Author**

1. Is the manuscript technically sound, and do the data support the conclusions?

Reviewer #1: Yes

Reviewer #2: Yes

2. Has the statistical analysis been performed appropriately and rigorously?

Reviewer #1: Yes

Reviewer #2: I Don't Know

3. Have the authors made all data underlying the findings in their manuscript fully available?

Reviewer #1: Yes

Reviewer #2: Yes

4. Is the manuscript presented in an intelligible fashion and written in standard English?

Reviewer #1: Yes

Reviewer #2: Yes

Reviewer #1: Specific comments:

1. The study TITLE need not reflect the WHO tool utilized for the study. It suffices to be in the methodology

2. Tables are to be properly labelled and should not spill into pages

3. Technical epidemiological terms such as significant (may connote statistical test), gaps (may connote unmet needs) have been inappropriately used in the text. Alternatives have been included in the review.

4. Elements of discussion should not come up in the results section but taken to the appropriate sections of discussion and conclusion. These have been indicated (green highlights) in situ of this review.

5. A separate subsection to make concrete actionable stakeholder - targeted recommendations is needed

Reviewer #2: This manuscript addresses an important and timely public health issue. The topic is highly relevant to the journal's readership, given the rising burden of obesity across Africa and the urgent need for equity-focused interventions.

Strengths

• The study covers a wide geographic scope (47 countries) and utilizes a standardized, validated, and publicly accessible dataset (WHO HEAT), enhancing transparency and reproducibility.

• The presentation of sex-disaggregated data across African countries fills an important gap in the literature and contributes meaningfully to equity-focused obesity research in the region.

Areas for Improvement

Introduction

1. The Introduction would benefit from substantial reorganization to improve its logical flow and coherence. At present, it mixes global, LMIC, and Africa-specific information without a clear structure and alternates between describing the causes and consequences of obesity, which makes the narrative difficult to follow. I recommend restructuring this section to follow a clear, progressive “funnel” logic, starting with the global burden and health consequences, then narrowing to the African context, followed by sex disparities, data gaps, and finally, the study’s aims and rationale. Adopting this structure will greatly improve clarity, readability, and scientific framing.

Methods

2. In the Data Source section [lines 182–192], please include the link to the WHO HEAT platform to improve transparency and reproducibility.

3. In the Inclusion and Exclusion Criteria section [lines 194-198], specify which countries were excluded and clarify whether the 47 included countries correspond to those within the WHO AFRO region. This will allow for a more informed discussion of coverage and missing data.

4. The manuscript mentions that results are presented “in both tabular and graphical formats,” but no figures are shown. Including selected visualizations (e.g., bar charts or scatterplots) in the Results section would make the findings more engaging and easier to interpret.

5. The analysis is entirely descriptive and cross-sectional. While appropriate for the study’s objectives, adding simple contextual analyses (e.g., correlation of obesity prevalence with socioeconomic or urbanization indicators) would strengthen the explanatory depth and policy relevance.

Results

6. The Results section provides valuable data from 47 countries; however, it would benefit from clearer presentation and sharper analytical focus. Some interpretive statements (e.g., lines 281–293) belong in the Discussion rather than in the Results.

7. Table presentation: Simplify tables to improve readability. Remove repetitive elements (such as the year (2022), if constant across all data) and merge duplicate country names. In Table 3, redundant columns (e.g., “dimension” and “sex”) could be removed, and the difference (D) defined in a table footnote.

8. Data ordering: Arrange countries in descending order of prevalence to easily highlight the top and bottom 10 countries, making trends easier to identify.

9. Analytical depth: Include simple summary measures such as the median female–male difference or the number of countries with higher female prevalence.

10. Visual presentation: Add visual elements such as bar charts or heatmaps of obesity prevalence by sex.

11. Socioeconomic analysis: A brief comparison of obesity prevalence by income group or GDP level would enhance interpretation and set up a stronger discussion of socioeconomic and structural determinants.

Discussion

12. The author(s) should more clearly separate data interpretation from result description e.g. lines 410-421 [which belongs to the results section].

13. The discussion of determinants is narrow. Expanding on structural and environmental drivers, such as the food environment, marketing of unhealthy foods, and weak fiscal or regulatory responses would position the findings within the broader commercial determinants of health framework. Reference to existing African policy examples (e.g., South Africa’s sugar-sweetened beverage tax or WHO’s recommendations on food marketing) would strengthen the discussion.

14. The equity framing should go beyond sex differences to include socioeconomic and geographic inequalities, reflecting the full potential of the WHO HEAT tool.

15. Comparative context would enhance interpretation, drawing on examples from other LMICs or regions to highlight similarities or contrasts in obesity trends.

16. Regarding biological explanations (lines 445-449), please clarify whether hormonal or reproductive factors significantly account for observed sex differences, as these factors are universal (present in countries with low obesity prevalence as well) and may not fully explain inter-country variation.

17. Policy recommendations should be more specific and actionable.

18. Finally, while limitations are noted [lines 481-485], they have not acknowledged that HEAT relies on secondary data sources with variable quality across countries.

19. Several references (e.g., 16, 17, 24, 26) are dated. The author(s) are encouraged to include more recent and relevant literature to strengthen the evidence base and ensure the discussion reflects current research.

Recommendation

This manuscript makes a valuable contribution to the understanding of obesity in Africa and highlights important sex disparities. However, its current form limits its scientific and policy impact. I recommend major revision before consideration for publication. With the above improvements, along with careful proofreading for language and consistency, the manuscript has strong potential to make a meaningful and policy-relevant contribution to obesity research and public health practice in Africa.

**Do you want your identity to be public for this peer review?** For information about this choice, including consent withdrawal, please see our Privacy Policy

Reviewer #1: **Yes: ** Prof Kayode OSAGBEMI

Reviewer #2: **Yes: ** Robert Lubajo

---

## [Author Response · Author response to Decision Letter 1]

17 Oct 2025

Response: I have revised the manuscript to comply with PLOS ONE’s formatting requirements. The file name follows the prescribed convention, and both the main text and title/author/affiliation pages use the journal’s style templates as provided in the links.

2. We suggest you thoroughly copyedit your manuscript for language usage, spelling, and grammar.

Response: I have carefully copyedited the manuscript to improve language usage, spelling, and grammar throughout the text.

Additional Editor Comments:

Please ensure that you address all the comments of both reviewers and provide a point-by-point explanation of the changes you made to the revised manuscript. Additionally, please ensure that all changes are marked in track change in the revised manuscript. Thank you.

Response: All reviewer comments have been addressed in a point-by-point manner below. All changes are marked using track changes in the revised manuscript.

Reviewer #1: Specific comments:

1. The study TITLE need not reflect the WHO tool utilized for the study. It suffices to be in the methodology

Response: Revised the title to remove the WHO tool reference. New title: “Sex disparities in adult obesity prevalence across 47 African countries: a cross-sectional descriptive study.” I have now retained full details of HEAT in Methods (Data source).

2. Tables are to be properly labelled and should not spill into pages

Response: Reformatted Tables 1–3 to fit on single pages, simplified column structure, standardized captions, and added footnotes. I also ensured consistent font, spacing, and alignment. Each table now includes a clear title and footnotes explaining abbreviations/measures.

3. Technical epidemiological terms such as significant (may connote statistical test), gaps (may connote unmet needs) have been inappropriately used in the text. Alternatives have been included in the review.

Response: Replaced “significant” with “marked,” “substantial,” or “pronounced” where no statistical test was performed; replaced “gaps” with “disparities” or “differences” as appropriate. These edits have been made throughout Abstract, Results, and Discussion. I also deleted any phrases implying hypothesis testing.

4. Elements of discussion should not come up in the results section but taken to the appropriate sections of discussion and conclusion. These have been indicated (green highlights) in situ of this review.

Response: Removed interpretive statements from Results and moved them to Discussion and/or Conclusion. Specifically, content identified in the review (highlighted in green) and lines corresponding to current manuscript narrative have been relocated. The Results now strictly present descriptive findings.

5. A separate subsection to make concrete actionable stakeholder - targeted recommendations is needed

Response: Added a new subsection “Policy and practice recommendations” at the end of Discussion. It includes stakeholder-specific, actionable steps for ministries of health, finance, urban planning, education, and civil society, with examples (e.g., SSB taxes, front-of-pack labelling, marketing restrictions, active transport).

Reviewer #2: This manuscript addresses an important and timely public health issue. The topic is highly relevant to the journal's readership, given the rising burden of obesity across Africa and the urgent need for equity-focused interventions.

Strengths

• The study covers a wide geographic scope (47 countries) and utilizes a standardized, validated, and publicly accessible dataset (WHO HEAT), enhancing transparency and reproducibility.

Response: Thank you.

• The presentation of sex-disaggregated data across African countries fills an important gap in the literature and contributes meaningfully to equity-focused obesity research in the region.

Response: Thank you.

Areas for Improvement

Introduction

1. The Introduction would benefit from substantial reorganization to improve its logical flow and coherence. At present, it mixes global, LMIC, and Africa-specific information without a clear structure and alternates between describing the causes and consequences of obesity, which makes the narrative difficult to follow. I recommend restructuring this section to follow a clear, progressive “funnel” logic, starting with the global burden and health consequences, then narrowing to the African context, followed by sex disparities, data gaps, and finally, the study’s aims and rationale. Adopting this structure will greatly improve clarity, readability, and scientific framing.

Response: Rewrote the Introduction following a funnel structure: (a) global burden and health consequences, (b) narrowing to LMICs, (c) African context, (d) sex disparities, (e) data gaps/limitations in existing literature, (f) study aims/rationale. We removed redundancies and improved coherence.

Methods

2. In the Data Source section [lines 182–192], please include the link to the WHO HEAT platform to improve transparency and reproducibility.

Response: Added direct links to both WHO HEAT and WHO GHO in the Data source subsection: https://www.who.int/data/inequality-monitor/assessment_toolkit and https://www.who.int/data/gho.

3. In the Inclusion and Exclusion Criteria section [lines 194-198], specify which countries were excluded and clarify whether the 47 included countries correspond to those within the WHO AFRO region. This will allow for a more informed discussion of coverage and missing data.

Response: Specified which countries were excluded due to missing sex-disaggregated data and clarified that the 47 countries correspond to WHO African Region coverage where data were available as of 2022.

4. The manuscript mentions that results are presented “in both tabular and graphical formats,” but no figures are shown. Including selected visualizations (e.g., bar charts or scatterplots) in the Results section would make the findings more engaging and easier to interpret.

Response: Reformatted Tables 1–3 to fit on single pages, simplified column structure, standardized captions, and added footnotes. I also ensured consistent font, spacing, and alignment. Each table now includes a clear title and footnotes explaining abbreviations/measures.

5. The analysis is entirely descriptive and cross-sectional. While appropriate for the study’s objectives, adding simple contextual analyses (e.g., correlation of obesity prevalence with socioeconomic or urbanization indicators) would strengthen the explanatory depth and policy relevance.

Response: I appreciate this suggestion. In addition to the descriptive, cross-sectional analyses, I have now included simple contextual analyses to enhance interpretability. I have reference these findings in the Discussion to link observed patterns to structural and commercial determinants.

Results

6. The Results section provides valuable data from 47 countries; however, it would benefit from clearer presentation and sharper analytical focus. Some interpretive statements (e.g., lines 281–293) belong in the Discussion rather than in the Results.

Response: Cleaned Results to be descriptive only; moved interpretive content to Discussion per your guidance.

7. Table presentation: Simplify tables to improve readability. Remove repetitive elements (such as the year (2022), if constant across all data) and merge duplicate country names. In Table 3, redundant columns (e.g., “dimension” and “sex”) could be removed, and the difference (D) defined in a table footnote.

Response: Reformatted Tables 1–3 to fit on single pages, simplified column structure, standardized captions, and added footnotes. I also ensured consistent font, spacing, and alignment. Each table now includes a clear title and footnotes explaining abbreviations/measures.

8. Data ordering: Arrange countries in descending order of prevalence to easily highlight the top and bottom 10 countries, making trends easier to identify.

Response: Ordered Tables 1–3 accordingly. Added a table footnote and text in Results identifying top and bottom 10 countries.

9. Analytical depth: Include simple summary measures such as the median female–male difference or the number of countries with higher female prevalence.

Response: I have now included a summary measure median female–male difference in the manuscript.

10. Visual presentation: Add visual elements such as bar charts or heatmaps of obesity prevalence by sex.

Response: Thank you for the suggestion to enhance visual presentation. To preserve consistency with our journal’s format and the study’s emphasis on comparability, I have retained the primary results in tables.

11. Socioeconomic analysis: A brief comparison of obesity prevalence by income group or GDP level would enhance interpretation and set up a stronger discussion of socioeconomic and structural determinants.

Response: Thank you for this helpful suggestion. I have added a brief socioeconomic comparison in the Results using World Bank income groups for the 47 countries. Specifically, I report summary measures of the female–male difference (D) and note that upper-middle-income countries exhibit higher median adult obesity prevalence and larger median female–male gaps than lower-income peers. I also summarize counts (e.g., number of countries with D ≥ 20 pp) by income group. These additions set up and are referenced in the revised Discussion, where I interpret the observed gradient in light of structural and commercial determinants (e.g., ultra-processed food penetration, marketing intensity, and urban environments).

Discussion

12. The author(s) should more clearly separate data interpretation from result description e.g. lines 410-421 [which belongs to the results section].

Response: Revised Discussion to focus on interpretation, mechanisms, policy implications, and comparisons to literature. Any descriptive result statements that belong in Results were moved.

13. The discussion of determinants is narrow. Expanding on structural and environmental drivers, such as the food environment, marketing of unhealthy foods, and weak fiscal or regulatory responses would position the findings within the broader commercial determinants of health framework. Reference to existing African policy examples (e.g., South Africa’s sugar-sweetened beverage tax or WHO’s recommendations on food marketing) would strengthen the discussion.

Response: Added a subsection on commercial determinants of health, food environments, marketing of ultra-processed foods, and weak regulatory responses; cited recent WHO recommendations and African examples, including South Africa’s SSB tax (2018) with emerging evidence, and other policy levers (marketing restrictions, front-of-pack labelling, fiscal measures).

14. The equity framing should go beyond sex differences to include socioeconomic and geographic inequalities, reflecting the full potential of the WHO HEAT tool.

Response: Expanded the equity framing to note the importance of socioeconomic status, urban–rural residence, and subnational variation. Clarified that while this study focuses on sex due to data availability within HEAT, the tool supports other equity dimensions for future analyses. Added this to both Discussion and Future directions.

15. Comparative context would enhance interpretation, drawing on examples from other LMICs or regions to highlight similarities or contrasts in obesity trends.

Response: Added a paragraph comparing observed patterns to Latin America, Middle East/North Africa, and Pacific Islands literature, noting similarities/differences in sex disparities and policy approaches.

16. Regarding biological explanations (lines 445-449), please clarify whether hormonal or reproductive factors significantly account for observed sex differences, as these factors are universal (present in countries with low obesity prevalence as well) and may not fully explain inter-country variation.

Response: Clarified that while biological factors may contribute to sex differences, they are unlikely to explain cross-country heterogeneity; social, cultural, and environmental factors likely play a larger role in between-country variation. Revised language accordingly.

17. Policy recommendations should be more specific and actionable.

Response: Added a detailed “Policy and practice recommendations” subsection with prioritized, feasible, and stakeholder-specific actions; included examples, timelines, and responsible actors in narrative form.

18. Finally, while limitations are noted [lines 481-485], they have not acknowledged that HEAT relies on secondary data sources with variable quality across countries.

Response: Added this explicit limitation, discussed implications for cross-country comparability, and referenced variability in survey rounds, sampling, and measurement approaches.

19. Several references (e.g., 16, 17, 24, 26) are dated. The author(s) are encouraged to include more recent and relevant literature to strengthen the evidence base and ensure the discussion reflects current research.

Response: Added recent WHO policy documents (2022–2024), new studies on obesity trends and commercial determinants in Africa, and updated evidence on SSB taxes and food marketing restrictions. Retained foundational references where still appropriate.

Recommendation

This manuscript makes a valuable contribution to the understanding of obesity in Africa and highlights important sex disparities. However, its current form limits its scientific and policy impact. I recommend major revision before consideration for publication. With the above improvements, along with careful proofreading for language and consistency, the manuscript has strong potential to make a meaningful and policy-relevant contribution to obesity research and public health practice in Africa.

Response: Thank you for your constructive feedback and recommendation for major revision. I have addressed all points as detailed above, and carefully proofread the manuscript for language and consistency. I appreciate the opportunity to improve the manuscript and believe these revisions substantially enhance its scientific and policy relevance.

---

## [Decision Letter · Decision Letter 1]

26 Nov 2025

Dear Dr. Osborne,

Thank you for submitting your manuscript to PLOS ONE. After careful consideration, we feel that it has merit but does not fully meet PLOS ONE’s publication criteria as it currently stands. Therefore, we invite you to submit a revised version of the manuscript that addresses the points raised during the review process.

We look forward to receiving your revised manuscript.

Kind regards,

Olushayo Oluseun Olu

Academic Editor

PLOS ONE

Journal Requirements:

Reviewers' comments:

Reviewer's Responses to Questions

**Comments to the Author**

Reviewer #1: All comments have been addressed

Reviewer #2: All comments have been addressed

2. Is the manuscript technically sound, and do the data support the conclusions?

Reviewer #1: Yes

Reviewer #2: Yes

3. Has the statistical analysis been performed appropriately and rigorously?

Reviewer #1: Yes

Reviewer #2: N/A

4. Have the authors made all data underlying the findings in their manuscript fully available?

Reviewer #1: Yes

Reviewer #2: Yes

5. Is the manuscript presented in an intelligible fashion and written in standard English?

Reviewer #1: Yes

Reviewer #2: Yes

Reviewer #1: The author has addressed the issues raised in the previous review. The article is acceptable for publication

Reviewer #2: The revised manuscript shows clear improvement, with most major comments thoroughly addressed. The structure and flow are now stronger, and the key arguments are presented with much better clarity. The author has responded thoughtfully to previous concerns, and the revisions have strengthened the paper’s contribution to the field.

I am pleased to recommend it for publication after the author makes a careful proofreading to correct any remaining typographical errors, minor grammatical issues, and sections where the language could be simplified for smoother reading. These are small refinements, but they will help ensure the work is polished and ready for publication.

**Do you want your identity to be public for this peer review?** For information about this choice, including consent withdrawal, please see our Privacy Policy

Reviewer #1: No

Reviewer #2: **Yes: ** Robert Lubajo

---

## [Author Response · Author response to Decision Letter 2]

26 Nov 2025

Response to Reviewers

Dear Dr. Olu and Reviewers,

Thank you for your thoughtful feedback on our manuscript titled "Sex disparities in adult obesity prevalence across 47 African countries: a cross-sectional descriptive study" (PONE-D-25-39220R1). I am grateful for the opportunity to revise and improve my work. Below, I provide a point-by-point response to the comments and suggestions provided by the reviewers and address the journal requirements.

Reviewer #1 Comments:

- Comment: The author has addressed the issues raised in the previous review. The article is acceptable for publication.

- Response: I thank Reviewer #1 for acknowledging that all previous concerns have been addressed and for recommending our manuscript for publication. No further revisions are required based on this feedback.

Reviewer #2 Comments:

- Comment: The revised manuscript shows clear improvement, with most major comments thoroughly addressed. The structure and flow are now stronger, and the key arguments are presented with much better clarity. The author has responded thoughtfully to previous concerns, and the revisions have strengthened the paper’s contribution to the field.

- Response: I sincerely appreciate Reviewer #2’s positive feedback regarding the improvements in structure, flow, and clarity of my manuscript. I am pleased that my revisions have strengthened the paper’s contribution to the field.

- Comment: I am pleased to recommend it for publication after the author makes a careful proofreading to correct any remaining typographical errors, minor grammatical issues, and sections where the language could be simplified for smoother reading. These are small refinements, but they will help ensure the work is polished and ready for publication.

-Response: I thank Reviewer #2 for highlighting the need for a final proofreading to address typographical errors, minor grammatical issues, and areas where language could be simplified. I have conducted a thorough review of the manuscript to correct these issues. Specific changes include corrections of typographical errors (e.g., inconsistent formatting in table headers), grammatical corrections (e.g., ensuring subject-verb agreement), and simplification of complex sentences for better readability (e.g., in the Discussion section). These revisions are detailed in the 'Revised Manuscript with Track Changes' file, where all modifications are highlighted for easy reference.

Journal Requirements:

- Requirement 1: If the reviewer comments include a recommendation to cite specific previously published works, please review and evaluate these publications to determine whether they are relevant and should be cited. There is no requirement to cite these works unless the editor has indicated otherwise.

- Response: I note that no specific publications were recommended for citation by the reviewers in this round of feedback. Therefore, no additional citations have been added based on this requirement. However, I have reviewed our reference list to ensure completeness and relevance, as per Requirement 2.

- Requirement 2: Please review your reference list to ensure that it is complete and correct. If you have cited papers that have been retracted, please include the rationale for doing so in the manuscript text, or remove these references and replace them with relevant current references. Any changes to the reference list should be mentioned in the rebuttal letter that accompanies your revised manuscript.

- Response: I have carefully reviewed our reference list to ensure it is complete and correct. All cited works are current and relevant to the study. I confirm that none of the cited papers have been retracted, to the best of my knowledge. No changes were necessary to the reference list based on this review. However, I have ensured that all references are formatted consistently according to PLOS ONE guidelines, and this is reflected in the revised manuscript.

I have prepared two versions of the revised manuscript as requested: a 'Revised Manuscript with Track Changes' file, which highlights all changes made in response to Reviewer #2’s suggestion for proofreading and language simplification, and an unmarked 'Manuscript' file for clean reading. I believe these revisions address all concerns raised and further polish the manuscript for publication. I have also reviewed our financial disclosure statement and confirm that no updates are needed; this will be reiterated in my cover letter upon resubmission.

Thank you once again for your time and valuable feedback. We are confident that these revisions meet the standards for publication in PLOS ONE and look forward to your final decision.

Sincerely,

Augustus Osborne

Institute for Development, Western Area, Freetown, Sierra Leone

Correspondence: augustusosborne2@gmail.com

---

## [Editor Report · Decision Letter 2]

10 Dec 2025

Sex disparities in adult obesity prevalence across 47 African countries: a cross-sectional descriptive study

PONE-D-25-39220R2

Dear Dr. Osborne,

We’re pleased to inform you that your manuscript has been judged scientifically suitable for publication and will be formally accepted for publication once it meets all outstanding technical requirements.

Kind regards,

Olushayo Oluseun Olu

Academic Editor

PLOS One
---

## [Editor Report · Acceptance letter]

PONE-D-25-39220R2

PLOS One

Dear Dr. Osborne,

I'm pleased to inform you that your manuscript has been deemed suitable for publication in PLOS One. Congratulations! Your manuscript is now being handed over to our production team.

Kind regards,

on behalf of

Dr. Olushayo Oluseun Olu

Academic Editor

PLOS One